# Collaborative Training of Tiny-Large Vision Language Models

Shichen Lu
sclu2020@buaa.edu.cn
School of Computer
Science and Engineering,
Beihang University
Beijing, China

Longteng Guo
longteng.guo@nlpr.ia.ac.cn
Institute of Automation,
Chinese Academy of
Sciences
School of Artificial
Intelligence, University of
Chinese Academy of
Sciences, Beijing, China

Wenxuan Wang
wangwenxuan2023@ia.ac.cn
Institute of Automation,
Chinese Academy of
Sciences
School of Artificial
Intelligence, University of
Chinese Academy of
Sciences, Beijing, China

Zijia Zhao
zhaozijia2021@ia.ac.cn
Institute of Automation,
Chinese Academy of
Sciences
School of Artificial
Intelligence, University of
Chinese Academy of
Sciences, Beijing, China

Tongtian Yue
yuetongtian2022@ia.ac.cn
Institute of Automation,
Chinese Academy of
Sciences
School of Artificial
Intelligence, University of
Chinese Academy of
Sciences, Beijing, China

Jing Liu[*]
jliu@nlpr.ia.ac.cn
Institute of Automation,
Chinese Academy of
Sciences
School of Artificial
Intelligence, University of
Chinese Academy of
Sciences, Beijing, China

Si Liu
liusi@buaa.edu.cn
Institute of Artificial
Intelligence, Beihang
University
Beijing, China

## Abstract

Recently, large vision language models (LVLMs) have advanced AI by integrating visual and linguistic data for tasks like visual conversation, image captioning, and visual question answering. Current LVLM research either scales up model size for performance or reduces parameters for limited computational resources. We believe both large and tiny models have unique strengths and that collaborative training yields better results than independent training. We propose Collaborative Training of Tiny-Large Vision Language Models (CTVLMs), a framework connecting large and tiny models via a projection layer and leveraging a synergistic training strategy. Our framework improves training efficiency by strengthening the interconnection between large and tiny models. Using the parameter efficiency of tiny models, we effectively align image-text features, then apply knowledge distillation to help large models better align cross-modal information. During fine-tuning, the large model's extensive knowledge enhances tiny model's performance. This collaborative approach allows models to adapt to various computational resources and outperforms existing methods in vision-language tasks.

[*]Corresponding Author.

## CCS Concepts

• **Computing methodologies** → **Computer vision**.

## Keywords

Multimodal, Large Language Model, Collaborative Training, Knowledge Distillation

### ACM Reference Format:

Shichen Lu, Longteng Guo, Wenxuan Wang, Zijia Zhao, Tongtian Yue, Jing Liu, and Si Liu. 2024. Collaborative Training of Tiny-Large Vision Language Models. In *Proceedings of the 32nd ACM International Conference on Multimedia (MM '24), October 28-November 1, 2024, Melbourne, VIC, Australia.* ACM, New York, NY, USA, 10 pages. https://doi.org/10.1145/3664647.3681026

## 1 Introduction

In recent years, multimodal large-scale pre-trained models have emerged as a major breakthrough in artificial intelligence, capturing significant research interest. By amalgamating diverse data types, including visual and linguistic information, these models[2, 9, 22, 31, 68, 77, 79, 80, 83, 86–88] exhibit a profound ability to understand and execute complex tasks such as image captioning and visual question answering. The advancement of these models has considerably pushed the frontiers of machine comprehension and generative abilities, establishing a foundation for enhanced human-computer interaction and more intelligent services.

The prevailing approach in Vision Large Language Models research is to scale up the model size and feed more data to achieve better scaling effects [2, 3, 7, 9, 13, 43, 51, 74]. Concurrently, as the model size scales up, the demand for computational resources increases, making the inference deployment of LVLMs less efficient and more challenging. Consequently, there has been a surge in

Shichen Lu, Longteng Guo, Wenxuan Wang, Zijia Zhao, Tongtian Yue, Jing Liu, and Si Liu

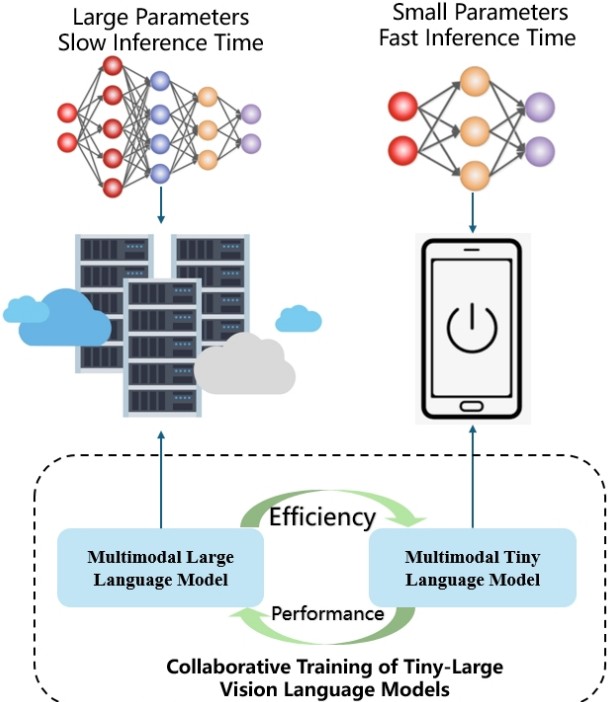

**Figure 1: Compared with previous visual language models, the independent training method could only produce two types of models separately. Large language models: They possess impressive performance but require substantial computational resources. Tiny language models: Require minimal computational resources, making them easy to deploy in environments with limited capabilities, such as mobile phones. In our proposed structure, we conduct joint training of both large and small models, enabling them to handle various scenarios effectively.**

research aimed at improving the efficient inference and ease of deployment for LVLMs. This includes studies on model compression, pruning, and distillation [11, 30, 63, 69] to address these challenges.

However, these two mainstream research directions are independent and lack integration. The current training strategy is relatively inefficient, demonstrates poor inter-model connectivity, and does not effectively leverage the respective strengths of large and tiny models. In other words, lacking a comprehensive approach that leverages the strengths of both to mutually enhance the training process. Therefore, we are inspired to ask: is it possible to jointly leverage both training regimes and develop a more effective training framework that simultaneously boosts the performance of both tiny and large MLLMs?

To this end, we introduce a novel collaborative training strategy for both large and tiny language models, aimed at simultaneously enhancing the performance of both through an efficient mutual learning approach. At the core of our collaborative training, the larger model can impart superior knowledge to the tiny model through knowledge distillation, while the tiny model can assist the large model in achieving multimodal alignment and provide a

broader range of multimodal information feature representations than a single model setup. Specifically, during the training process of our proposed framework, we employed CLIP-VIT-L/14[52] as the image encoder and OPT-125m[90] as the tiny language model for the text encoder, which is a representative and lightweight language model. Additionally, Vicuna-7B was selected as the Large Language Model, we use two linear layers to interconnect each other.

Following our proposed framework, the model underwent a two-stage training process: alignment and instruct tuning. In the first phase, the tiny language model-led guide of the large language model. We use a large corpus of image-text pairs was utilized to align the token embedding and multimodal features. Compared to directly aligning with the large language model, the tiny language model has a similar parameter count to the image encoder, facilitating convenient and effective multimodal alignment. After alignment the image-text feature between vision model and tiny model, we employ knowledge distillation techniques[21], we use the powerful multimodal image-text capabilities of the tiny model to enhance the cross-modal information alignment in the large model. In the tuning phase, the rich knowledge reservoir of the large language model was utilized to enhance the multimodal understanding capabilities of the tiny model. It is noteworthy that we use reverse Kullback-Leibler Divergence (KLD)[18] to guide the learning of tiny model by large model, it encourages the student to generate samples preferred by the teacher within its own capacities. During the training phase, we used the feature outputs from the tiny model to replace the previously used unimodal image feature in the large language model. Incorporating both image and text features enabled more effective multimodal fusion, enhancing the large language model's ability to understand image-text relationships.

In summary, our proposed collaborative training approach rapidly aligns the image encoder with the tiny language model. Subsequently, we employ a distillation strategy to help the large model align cross-modal information. During the fine-tuning phase, the multimodal features output by the tiny model enhance the large model's multimodal understanding capabilities. Furthermore, the large model can impart a richer knowledge base to the tiny model.

Our main contributions can be summarized as follows:

- We propose a novel approach called Collaborative Training of Tiny-Large Vision Language Models (CTVLMs) that enhances the alignment of multimodal information between large and tiny models. This method leverages the efficiency of the tiny model to facilitate a more effective alignment process. The tiny model not only improves multimodal alignment but also enriches the large model's understanding of image-text relationships.

- We propose a novel collaborative training strategy where the large model imparts its extensive knowledge to the tiny model, optimizing the learning process. This improves the tiny model's performance and enhances the large model's ability to handle cross-modal information. This mutual learning approach significantly advances both models beyond traditional single-model training methods.

- The experimental results from the vision-language and multimodal dialogue benchmarks convincingly demonstrate the superiority of our collaborative training strategy over previous state-of-the-art methods.

## 2 Related Work

### 2.1 Large Vision Language Models

The integration of computer vision and natural language processing has led to the development of Vision-Language Models (VLMs), which combine visual and linguistic elements to enhance cross-modal comprehension and reasoning. This integration is pivotal in improving tasks that require both visual perception and language understanding. Inspired by the success of language models like BERT [29] and Transformer [71], various approaches, including VLBERT [64], UNITER [10], LXMERT [67], and ALBEF [38], have been developed to combine vision and language models, aiming to establish alignments between the two modalities. Additionally, within frameworks akin to the transformer-decoder model, such as OPT [90] and GPT [54], various methods employ a vision encoder to extract visual features, including GIT [72], SimVLM [78], CoCa [81], . These features are then utilized by a language decoder to generate textual responses based on the extracted visual information.

Recently, we have witnessed a rapid development on large language models [49, 70]. These models show superior performance over previous language models. These models, however, are characterized by their larger size, requiring significantly more computational resources. Recent developments on large language models have led to the emergence of Large Vision Language Models (LVLMs), designed to enhance language models with the capability to analyze and interpret visual information. For example, Flamingo [2] utilizes both visual and linguistic inputs to exhibit remarkable few-shot learning abilities in visual question-answering tasks. BLIP-2 [35] introduces a Q-former architecture that compresses vision features into a fixed number of queries for processing by large language models. Similarly, LLaVA [42] employs a straightforward linear layer to connect the vision encoder with the large language model. InstructBlip [13] further refines this approach by integrating instructions into the Q-former, allowing for the generation of instruction-specific visual queries. With the increasing capabilities of Vision Large Language Models (LVLMs), there has been a rapid growth in training data and parameters, escalating from initial 1B models to current systems with 34B and even exceeding 100B training parameters. This escalation significantly amplifies the demand for computational resources, slows down the inference process, and complicates deployment.

In this paper, we introduce a collaborative learning framework that concurrently optimizes both a tiny vision-language model and its larger counterpart. Within this framework, the smaller model is designed to extract more precise visual features, thereby enhancing the performance of the larger model. In turn, the larger model offers expert guidance to refine the capabilities of the tiny model.

### 2.2 Knowledge Distillation

Knowledge distillation[21], as a widely used model compression technique, aims at training a student model with the guidance of a teacher model [16, 56, 58]. In the NLP community, many works apply KD to text classification tasks by mimicking the teacher model's output distribution[40, 63, 89], hidden states[28, 66], or attention scores[73, 76]. For text generation, the standard KD method is to approximately minimize the forward KLD between the student's and the teacher's generation distribution by using the teacher's

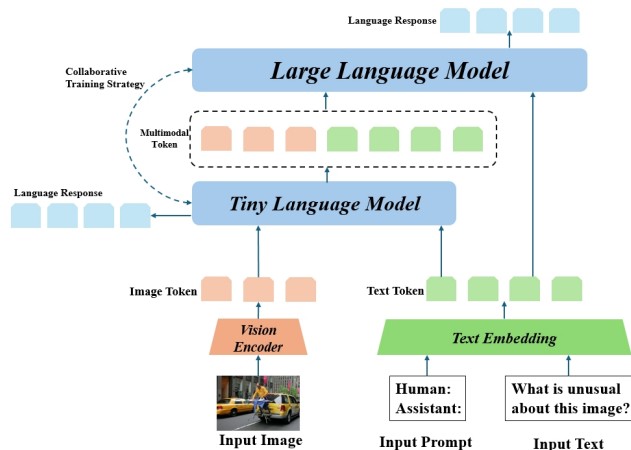

**Figure 2: The Architecture of our Collaborative Training of Tiny-Large Vision Language Models(CTVLMs). There are Three models in our Method, Vision Model, Tiny Language Model, and Large Language Model. First, we input the images into the Vision Encoder and Prompt connect with Text into the Text Embedding, resulting in separate image tokens and text tokens. We input both image and text tokens into Tiny Language Model to alignment visual and language features. Then the multimodal Tokens and Text Tokens input into the Large Language Model. Both our Tiny and Large Models can output Language Responses to achieve vision-language tasks. We use Collaborative Training Strategy to connect Tiny and Large language models.**

output at each time step as supervision[58] or direct training on the teacher's generated texts.

Thanks to the advancements in Large Language Models (LLMs), the current approach to knowledge distillation in LLMs has evolved beyond mere architectural compression. Some methods [19, 27, 69, 85] now emphasize a more sophisticated process of eliciting and transferring deep and expansive knowledge from large language models.

In this paper, we implement a reciprocal teaching strategy in which large and small models act as mutual educators. By employing a collaborative training framework, we aim to achieve enhanced performance for both models.

## 3 Method

### 3.1 Collaborative Architecture

As depicted in Figure2, unlike traditional vision-only backbones[20, 45, 75] and dual-encoder models[25, 53, 65], the architecture of our CTVLMs comprises a vision encoder, two closely integrated language models with different magnitudes. The large and tiny models are mirrors of each other, differing mainly in parameter count, allowing for a unified infrastructure and optimization. We use two MLP projection layers to map modality features into the LLM semantic space, with Vicuna-7B[11] for the large model and OPT-125m[90] for the tiny model. This setup validates our collaborative training approach, harmonizing small and large models.

Our progressive strategy begins with pre-training the tiny model to synchronize vision and language modalities through generative learning, followed by comprehensive collaborative training using Knowledge Distillation. This training is divided into two phases: feature alignment and instruct tuning. In the feature alignment phase, the tiny model rapidly aligns visual and linguistic features. Distillation techniques then help the large model align. During tuning, the large model enriches the tiny model's knowledge through reverse distillation.

**Tiny Language Model:** We implement the tiny language model of our method with vanilla OPT[90]. To match the scale of the vision encoder, we choose OPT-125m[90] as our tiny language model(Tiny LMs) with only 115 million trainable parameters. We use a tiny language model to align visual and linguistic features. As shown in Figure2, Tiny LMs is developed based on the pre-trained OPT-125m and exchanges the token embeds as Vicuna[11] and newly added cross attention layers. This manner allows Tiny LMs to smoothly integrate visual elements into the language model, thereby enhancing the coherence and effectiveness of the combined features.

**Collaborative Model:** Both our tiny and large LMs can support various vision-language tasks. Given an input image $I \in \mathbb{R}^{H \times W \times 3}$, and input text $T$. Our model can generate a feature map $F_I$ and $F_T$, the feature feeds into large LMs as the reorganized visual representations from Tiny LMs. The subsequent text tokens are generated one by one sequentially. Compared to recently popular approaches, our method has two advantages: (1) Enhanced the interconnectedness of large and tiny LMs. (2) Comprehensively improving the performance of both large and tiny LMs.

We discuss the collaborative training based on Knowledge Distillation in Section 3.2. Subsequently, Section3.3 outlines the collaborative training strategy of our method.

## 3.2 Knowledge Distillation for Collaborative training

In our collaborative training framework, we employ distillation and reverse distillation techniques to connect the large and tiny language models, allowing them to mutually learn from each other's strengths and thereby enhance each other's performance. We consider conditional text generation where the model produces a response $\boldsymbol{y} = \{y_t\}_{t=1}^{T}$ conditioning on a prompt $x$ sampled from the distribution $p_x$, which is typically how LLMs perform tasks. We formulate KD as an optimization problem to minimize the difference between a fixed teacher model distribution $p(\boldsymbol{y} \mid \boldsymbol{x})$ and a student model distribution $q_\theta(\boldsymbol{y} \mid \boldsymbol{x})$ parameterized by $\theta$. The standard KD methods approximately minimize the forward KLD:

$$\text{KL}\,[p\|q_\theta] = \mathbb{E}_{\boldsymbol{x} \sim p_{\boldsymbol{x}}, \boldsymbol{y} \sim p'} \log \frac{p(\boldsymbol{y} \mid \boldsymbol{x})}{q_\theta(\boldsymbol{y} \mid \boldsymbol{x})} \tag{1}$$

where $p'$ can be real data distribution (word-level KD) or teacher distribution $p$ (sequence-level KD). Though widely used, KL $[p\|q_\theta]$ tends to overestimate the void regions of $p$ in text generation tasks when $q_\theta$ is insufficiently expressive [26].

However, knowledge distillation (KD) is particularly suitable for large language models (LLMs) as they perform generative tasks, where the low-capacity student models struggle to replicate the

complex text generation patterns of teacher models or humans. To mitigate these challenges for the student model, we employ Reverse KLD in Stage Two. The learning objective for the LLM involves minimizing the reverse KLD between the student and teacher model distributions

$$\theta = \arg\min_\theta \mathcal{L}(\theta) = \arg\min_\theta \text{KL}\,[q_\theta\|p]$$
$$= \arg\min_\theta \left[ - \mathbb{E}_{\boldsymbol{x} \sim p_{\boldsymbol{x}}, \boldsymbol{y} \sim q_\theta} \log \frac{p(\boldsymbol{y} \mid \boldsymbol{x})}{q_\theta(\boldsymbol{y} \mid \boldsymbol{x})} \right] \tag{2}$$

Minimizing reverse KLD has been identified to induce mode-seeking behavior in generative modeling, where the model $q_\theta$ assigns high probabilities to the significant modes of $p$ while overlooking the smaller ones. This phenomenon is documented in studies like[8, 24, 33, 48]. On the contrary, minimizing forward KLD causes $q_\theta$ to allocate substantial probability masses to regions of $p$ with zero probability, leading to the production of low-quality text in practical applications. Reverse KLD, however, prioritizes the major modes of $p$, playing a vital role in ensuring the accuracy and reliability of text generation. Unlike sequence-level knowledge distillation that minimizes forward KLD, as seen in [30, 69], minimizing reverse KLD does not compel $q_\theta$ to conform to all $y$ sampled from the teacher distribution $p$. Instead, it prompts the student model to generate samples that align with the teacher's preferences within its capacity, a goal more attainable for the tiny language model.

## 3.3 Collaborative Training Strategy

**Stage I: Feature Alignment From Tiny to Large.(Tiny for Large)** Initially, we undertake generative learning to align the OPT125m with CLIP-vit, focusing on image-text pair embeddings. On the one hand, we can use the tiny language model to align image-text pairs more efficiently due to their comparable parameter scale. On the other hand, since they share the same token embeddings, we can also utilize distillation to better facilitate cross-modal alignment for the large language model. The publicly available LAION-400M dataset provides 80 million cleaned data samples, as summarized in Table1. During training, the text is encoded using OPT125m as $T_f$, and visual features are extracted with CLIP-vit as $I_f$.

According to BLIP-2's objective function, we compute the loss as the sum of image-text contrastive (ITC), image-text matching (ITM), and image-grounded text generation (ITG) losses. This process allows for the extraction of powerful multi-modal representations, aligning the feature space more closely with the Vision encoder. Unlike QFormer [35], our tiny language model inherently possesses promising image captioning abilities. Text tokens are sequentially generated, guiding the large language model with the Tiny language model. The tiny language model acquires robust multimodal capabilities through its initial alignment with the image encoder. Despite its strong linguistic abilities, the large language model's limited cross-modal alignment results in multimodal capabilities that lag behind the tiny model. Consequently, we choose to guide the large language model using the tiny language model. By leveraging the tiny language model's cross-modal alignment from the first stage, we aim to enhance the large language model's image-text comprehension capabilities. In the training phase, forward distillation, as shown in Formula 1, facilitates the tiny language

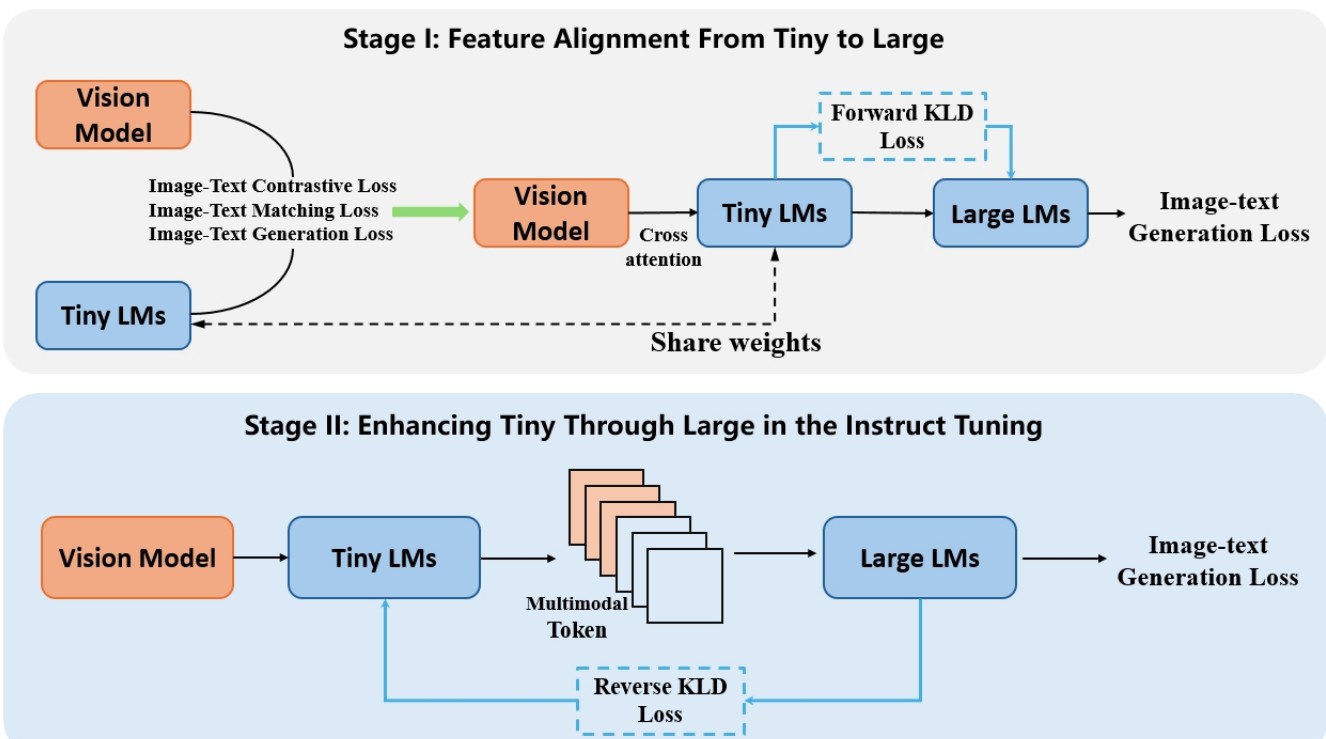

Figure 3: The Collaborative Training Strategy. It consists of two progressive stages, including Feature Alignment From Tiny to Large and Enhancing Tiny Through Large in the Instruct Tuning. In the First stage effectively leverage public data from diverse sources to align visual and Language features. In particular, we use Forwrd KLD to enhance the alignment capabilities of the Large Model. In the second stage we use Reverse KLD to enhance the performance of the Tiny Model. Both Tiny and Large Models have great performance in vision-language tasks.

model in guiding the large language model's representation learning. Probability distribution guidance supervision enables the large language model to effectively absorb the tiny language model's multimodal understanding capabilities. Concurrently, we freeze the large language model's parameters to prevent catastrophic forgetting. Notably, at this stage, only the projector inserted in the large MLLM is trained. The total loss supervision at this stage comprises caption loss and forward KLD loss, as mathematically expressed below

$$Loss_{total} = (1 - \lambda)Loss_{cap} + \lambda Loss_{kl} \qquad (3)$$

where $Loss_{total}$ representation the total loss including caption loss $Loss_{cap}$ and forward KLD loss: $Loss_{kl}$. Caption including both tiny and large language model. $\lambda$ is a balancing coefficient and in this stage $\lambda = 0.3$.

**Stage II: Enhancing Tiny through Large in Instruction Tuning.(Large for Tiny)** In this phase, we capitalize on the extensive knowledge reservoir of the large language model to steer the tiny model's learning process. large language models, with their capacity for developing nuanced feature representations, considerably enhance performance in specialized tasks. Their robustness, fostered by extensive exposure to varied data and complex structures, provides a strong defense against input noise and data disturbances.

Through the distillation of large language models into their smaller equivalents, we facilitate the transfer of these advanced capabilities, endowing the tiny model with a concentrated and potent knowledge base. This not only boosts their efficiency but also significantly elevates their effectiveness within their operational domain.

After the alignment in the first stage, both our tiny and large language models have developed exceptional multimodal understanding capabilities. For the large language model, we utilize the output of the tiny language model as input. The tiny model's output, enriched with combined image and text information, offers a richer cross-modal information set than traditional unimodal features, enhancing cross-modal knowledge comprehension. Subsequently, the large language model is employed to direct the learning process of the tiny model. In section 3.2, we explored two distinct distillation strategies, acknowledging that low-capacity student models struggle to replicate the complex text generation patterns of teacher models or humans. Thus, we implemented the reverse KLD algorithm for distilling the tiny model, detailed mathematically in 2. The supervision of total loss at this stage encompasses both caption and reverse KLD losses, mathematically outlined as follows:

$$Loss_{total} = (1 - \lambda)Loss_{cap} + \lambda Loss_{rekl} \qquad (4)$$

where $Loss_{total}$ representation the total loss including caption loss $Loss_{cap}$ and reverse KLD loss: $Loss_{rekl}$. Caption including both tiny and large language model.$\lambda$ is a balancing coefficient and in this stage $\lambda = 0.3$.

In summary, through our collaborative training approach, the large language model benefits from the tiny language model's generation of more detailed and informative multimodal semantic features, while the tiny language model benefits from the guidance provided by the large language model's knowledge, thus helping each other to improve performance. Additionally, our tiny language model can be deployed directly for inference without the need for new fine-tuning training for each task, as was required with previous middleware approaches.

## 4 Experiments

In this section, we empirically investigate the effectiveness of our CTVLMs method. We first provide the implementation and training details, then report the evaluation results on vision-language benchmarks compared with strong baselines. Then we provide analytic results for a better understanding of the model.

### 4.1 Implementation Details

**Model Configuration.** In this work, we built our CTVLMs framework following the implementation of LLaVA-1.5[42]. Specifically, we use the CLIP-VIT-L/14[52] as the Vision encoder(default resolution $336 \times 336$), OPT-125m[90] as the Tiny Language Model, and Vicuna-7B as the Large Language Model. After training, we designated the Tiny Language Model as CT-TinyLM and the Large Language Model as CT-LargeLM. We adopt AdamW [46] as the optimizer. For efficiency in training, we utilize DeepSpeed Zero Optimization stage 2. For stage one, we trained on 8 NVIDIA A800 GPUs using a batch size of 256 and training duration approximating 40 hours. Similarly, in stage two, we trained on 8 NVIDIA A800 GPUs using a batch size of 64 and training duration approximating 24 hours.

**Data Settings.** As shown in Table , we follow BLIP-2[36] to align the vision encoder with the tiny language model on LAION-400M[59] and CC3M[60] which filter our some extremely low-quality data to train our model. Then we follow the LLaVA-1.5[42] to align the multi-modal feature between tiny and large language models. Subsequently, we leverage the instruction dataset assembled from various sources to equip the model with a broad spectrum of capabilities.

**Feature Alignment From Tiny to Large.** Specifically, for more effective collaborative training in subsequent phases, we replaced the original token embedding layer with the one from LLaMA, identical to Vicuna's. The projection layer is randomly initialized[4]. The parameters in the tiny language model and projector layer are trainable. Then, the tiny Language model inherits its weights from before, while the new learnable projector layer between tiny and large language models is randomly initialized. Benefiting from the powerful representations learned in the numerous image-text pairs, we use the tiny language model as the teacher model to guide the large language model in better aligning between different modalities. Because the output feature of the tiny language model includes both image and text information, our method allows the

**Table 1: Datasets used for collaborative training. Among them, LAION-400M is web-scale image-text pairs data. CC3M is an academic caption dataset. LLaVA-1.5-558K and LLaVA-1.5-mix-665K are proposed and made public by LLAVA 1.5, used in alignment and visual instruction tuning.**

| Usage | Source | #Sample |
|-------|--------|---------|
| Stage I | LAION-400M[59], CC3M[6] LLaVA 1.5-558k[42] | 93M |
| Stage II | LLaVA 1.5-mix-665k[42] Single-turn Data | 1.5M |

large language model to obtain semantically stronger features with stronger correlations to understand images and text compared to using only image features. We keep both visual and language models frozen and only train the new projector's parameters.

**Enhancing Tiny through Large in Instruction Tuning.** We employ the large language model as the teacher model and utilize the reverse KLD method to distill the tiny language model. Because both our tiny and large language models have acquired strong multimodal understanding capabilities during Stage Two training, the substantial knowledge reservoir of the large language model becomes indispensable at this point. During fine-tuning, we enable the large language model to guide the tiny language model to acquire even more powerful capabilities. All parameters except the vision encoder are fully trainable.

### 4.2 Modality-Specific Evaluation

As illustrated in Table2, our model's intrinsic capabilities were assessed on the visual question-answering task, encompassing VQAv2[4], GQA[4], SQA[47], and TextVQA[62]. Beyond conventional multimodal tasks, the advent of ChatGPT [50] has shifted attention towards evaluating multimodal models in practical scenarios, particularly in multimodal dialogue. We tested the InternVL-Chat models on notable multimodal dialogue benchmarks such as MMMU [84], MME [84], MMB [84], SEED [34], and POPE [39]. MMMU evaluates comprehensive model understanding across various sensory inputs. MME offers a robust benchmark with 14 subtasks geared towards assessing the model's perceptual and cognitive abilities. MMB examines a broad spectrum of multimodal functionalities, including image-text matching and video captioning. SEED focuses on the quality of explanatory dialogue, while POPE serves as a prevalent dataset for evaluating object hallucination phenomena. For fairness, we report the activated parameters and image resolution of each model.

In Table3, we observe that with the assistance of the small model, the large model achieves an improved CIDEr score on the COCO-caption dataset during the alignment phase. This indicates that the small model, using distillation training strategies, enables better alignment of visual-language features in the large model. In Table 2, we compare the performance of models with and without the collaborative training strategy. It is evident that the performance of models subjected to distillation far surpasses those without it. Merely using the tiny language model as an intermediary module to align visual and linguistic features, similar to the Qformer in BLIP2,

**Table 2: Comparison with SOTA models on vision-language tasks. Vision question-answering VQAv2[17], GQA[23], SQA[47], TextVQA[62]. Multi-Modal Dialogue Benchmarks, including MMMU [84]; MME: the Perception and Cognition split sum score of MME [15]; MMB: MMBench [44]; SEED: SEED-Bench [34]; POPE [39]; The number reported in MMMU denote the performance on the val split. KLD mean Kullback-Leibler Divergence**

| Model | Res. | Act. | VQAv2 | GQA | SQA | TextVQA | MMMU | MME | MMB | SEED | POPE |
|---|---|---|---|---|---|---|---|---|---|---|---|
| BLIP-2 [36] | 224 | 7B | 41.0 | 41.0 | 61.0 | 42.5 | – | 1293.8 | – | 46.4 | 85.3 |
| InstructBLIP-7B [14] | 224 | 8.0B | – | 49.2 | – | 50.1 | 30.6 | 1391.4 | 36.0 | 53.4 | – |
| IDEFICS-9B [32] | 224 | 9B | 50.9 | 38.4 | – | 25.9 | – | – | 48.2 | – | – |
| Qwen-VL [3] | 448 | 9.6B | 78.8 | 59.3 | 67.1 | **63.8** | – | – | 38.2 | 56.3 | – |
| Qwen-VL-Chat [3] | 448 | 9.6B | 78.2 | 57.5 | 68.2 | 61.5 | **37.0** | 1487.5 | 60.6 | 58.2 | – |
| Fuyu-8B [5] | 1024 | 8B | 74.2 | – | – | – | – | 728.6 | 10.7 | – | 74.1 |
| LLaVA-1.5-7B [41] | 336 | 7.2B | 78.5 | 62.0 | 66.8 | 58.2 | – | 1510.7 | 64.3 | 58.6 | 85.9 |
| CT-LargeLM 7B w/o KLD | 336 | 7.4B | 78.9 | 62.5 | 67.5 | 60.3 | 34.1 | 1517.2 | 64.7 | 58.4 | 86.3 |
| CT-LargeLM 7B | 336 | 7.4B | **79.9** | **63.8** | **68.5** | 62.3 | 35.6 | **1540.2** | **67.4** | **59.2** | **86.6** |

**Table 3: Ablation study about the Tiny For Large demonstrates the multimodal alignment capabilities of our models. The CIDEr source in MS-COCO caption datasets. LLaVA1.5 is only trained with Stage 1.**

| Method | CIDEr |
|---|---|
| DALL-E [55] | 20.2 |
| ruDALL-E-XL[61] | 38.7 |
| minDALL-E[57] | 48.0 |
| X-LXMERT [12] | 55.8 |
| Parti[82] | 83.9 |
| Flamingo(3B;4-shot)[2] | 85.0 |
| Vanilla CM3[1] | 71.9 |
| LLaVA-1.5-7B(after the first training stage)[42] | 95.7 |
| CT-TinyLM | **98.3** |
| CT-LargeLM(w/o KLD) | 96.2 |
| CT-LargeLM(w KLD) | 97.6 |

where only increasing the MLP layer's parameters is not sufficient for better multimodal understanding. However, with our collaborative training strategy, the large language model significantly enhances its capabilities in visual-language tasks. At the scale of 7B activation parameters in table2 our LLM comprehensively surpasses LLaVA-1.5-7B. Moreover, it even exceeds Qwen-VL and Fuyu with 8B and 9B activated parameters. In Table3.2, compared to the strategy of training small models independently, our collaborative training strategy, leveraging knowledge from the large model, results in substantial improvements in visual-language tasks for tiny LMs. These experiments collectively validate the effectiveness of our method.

## 4.3 Ablation Study

In this section, we aim to illustrate the impact of distillation learning on enhancing mutual performance within our training framework. We investigate whether the tiny language model's fused features can acquire more comprehensive cross-modal information than a solely visual model. Our exploration includes evaluating the tiny language model's guidance on the large language model via distillation loss in the initial phase, and vice versa in the subsequent phase. Additionally, we assess whether the multimodal features extracted by the tiny language model can enhance the large model's understanding of multimodal information.

**Effect of Tiny for Large.** To investigate the tiny language model's instructive role for the large language model, we employed forward Knowledge Distillation (KLD) during the second phase to align the probability distribution output of the large language model with that of the tiny language model. As shown in Table3, after the first phase of training, LLaVA-1.5[42] achieved a CIDEr score of 95.7 on the coco-caption dataset. However, after pre-training on a large dataset, our tiny language model surpassed the performance of many 1B and larger models with a CIDEr score of 98.3 on coco-caption, demonstrating its strong multimodal alignment capability. Thus, through the distillation learning in the first phase, our large language model's abilities were also enhanced, proving the effectiveness of our method.

**Effect of Large for Tiny.** To validate the guiding role of the large language model on the tiny language model, in the second phase, we employed the reverse Knowledge Distillation (reKLD) algorithm to leverage the powerful knowledge capabilities of the large language model to distill the tiny model. We trained the tiny language model (only-opt) without the large language model. In Table4, we observed that without distillation, the large language model faces difficulties in effectively aiding the tiny language model to boost its performance. Moreover the reverse KLD yields better results compared to forward distillation. This might be due to the limited number of parameters in the tiny language model, which hinders its ability to learn the knowledge from the Large model fully. Instead, it encourages the student model to generate samples preferred by the teacher within its capacities, which improves the student model's performance after training, thereby maximizing the potential of our proposed training framework. Furthermore, the reverse KLD method enhances the selective learning capability of the tiny model, allowing it to focus on acquiring the most relevant and impactful knowledge from the large model. This method promotes a more efficient learning process and helps in overcoming

**Table 4: Ablation study about the Large For Tiny. The performance in vision question-answering Task. And also explores different distillation methods, demonstrating that reverse distillation yields better results.**

|  | VQAv2 | GQA | TextVQA |
|---|---|---|---|
| BLIP-2 OPT-6.7B [37] | 54.3 | 41.0 | 42.5 |
| Instruct-BLIP [14] | – | 49.2 | **50.1** |
| Only TinyLMs | 60.4 | 45.6 | 39.1 |
| TinyLMs(w/o KLD) | 61.5 | 46.3 | 40.2 |
| TinyLMs(w KLD) | 66.8 | 47.6 | 43.1 |
| TinyLMs(w reKLD) | **68.3** | **49.3** | 45.2 |

**Table 5: Ablation study about Multimodal Comprehension Enhancement. P represents prompt features, I for image features, and T for text features. And discussed the impact of different outputs on the performance of our model.**

|  |  | VQAv2 | GQA | Text VQA |
|---|---|---|---|---|
| TinyLMs | I+T | **68.3** | **49.3** | **45.2** |
|  | P+I+T | 68.2 | 49.1 | 45.1 |
|  | I | 67.9 | 49.1 | 45.0 |
| LargeLMs | I+T | **79.9** | **63.8** | **62.3** |
|  | P+I+T | 79.3 | 63.3 | 61.7 |
|  | I | 78.6 | 62.3 | 59.2 |

**Table 6: Ablation study of the proportion of KL-divergence Distillation Loss, we compare both Tiny and Large models' performance in vision-language tasks with different $\lambda$**

|  | $\lambda$ | VQAv2 | GQA | Text VQA |
|---|---|---|---|---|
| TinyLMs | 0.1 | 59.8 | 43.5 | 38.7 |
|  | 0.3 | **68.3** | **49.3** | **45.2** |
|  | 0.7 | 40.3 | 25.6 | 23.4 |
| LargeLMs | 0.1 | 78.9 | 62.7 | 60.5 |
|  | 0.3 | **79.9** | **63.8** | **62.3** |
|  | 0.7 | 63.8 | 54.9 | 53.1 |

the challenges associated with the parameter disparity between the two models.

**Multimodal Comprehension Enhancement.** To verify that integrating multimodal information in the input enhances the multimodal comprehension capability of large language models, compared to unimodal input, in the fine-tuning phase, previous large vision language models (LVLMs) typically mapped image features to text space through a simple linear layer before fusing them with textual information. In our approach, we utilize a tiny language model to merge image and text features, substituting the sole image feature input into the large language model, thereby endowing it with superior multimodal features and enhancing its capability in image-text understanding. In Table5, "P" represents prompt features, "I" for image features, and "T" for text features. Experiments reveal that the fused information of image and text substantially aids the large language model in image-text comprehension tasks. Meanwhile, since "P" is consistently the same and contains no relevant information, it proves to be ineffective during training. For the tiny language model, although the enhanced capabilities of the large language model could improve the tiny language model's performance through distillation, the decisive factor remains its own supervisory information, hence the output features from TinyLM itself have no impact on its performance.

**Effect of the proportion of Distillation.** The proportion of distillation loss significantly impacts model performance during the knowledge distillation process. Properly calibrating this loss proportion is crucial for achieving the optimal balance between knowledge transfer and model training efficiency. In this experiment, we explored the impact of the distillation ratio on the performance of our model. As shown in Table6, we observed When $\lambda = 0.7$ tiny LLMs get low performance in vision-language tasks. High distillation loss weight may lead to overfitting on the large model's specific behaviors, potentially ignoring the intrinsic learning capabilities of the tiny model. Due to their mutual influence, the large model ends up with suboptimal features, leading both models into a vicious cycle where their performance simultaneously deteriorates. When $\lambda = 0.1$, tiny LLMs might not sufficiently capture the nuanced knowledge or expertise from the Large model, resulting in suboptimal performance. Through experimentationwe use $\lambda = 0.3$ during our training process.

## 5 Conclusion

In our research, we explored the effectiveness and feasibility of joint training of large and tiny models, proposing a new Collaborative Training of Tiny-Large Vision Language Models(CTVLMs) framework that transforms the conventional strategy of independently training these models. Our framework integrates an image encoder with a closely connected pair of large and tiny models, enabling efficient alignment across modalities by leveraging the tiny model's superior capabilities in addressing cross-modal challenges. During the fine-tuning phase, we replace traditional image features with the multimodal information output from the tiny model, thereby providing the large model with a richer input of features. Simultaneously, we harness the large model's robust knowledge base to boost the overall performance of the tiny model. This approach not only enhances the efficiency of the training process but also significantly improves the performance and connectivity between the models. Our collaborative training strategy uses knowledge distillation to facilitate a robust exchange of capabilities, where the large model enriches the tiny model with its extensive knowledge, and the tiny model contributes to refining the multimodal comprehension of the large model. The strategic use of knowledge distillation and multimodal alignment allows for a more dynamic interaction between the models. This two-phase training process—alignment and instructive tuning—ensures that each model leverages the strengths of the other, leading to a more cohesive and potent learning system. This innovative methodology not only optimizes model performance but is also efficiently utilized.

## Acknowledgments

This research is supported in part by National Science and Technology Major Project (2022ZD0115502), National Natural Science Foundation of China (NO. 62122010, U23B2010), Zhejiang Provincial Natural Science Foundation of China (Grant No. LDT23F02022F02), Beijing Natural Science Foundation (NO. L231011), Beihang World TOP University Cooperation Program and Key Research and Development Program of Jiangsu Province under (Grant BE2023016-3).

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
