# OpenReview forum: "Collaborative Training of Tiny-Large Vision Language Models"
_acmmm.org/ACMMM/2024/Conference — MM2024 Poster_

### Official Review · Reviewer_n5za · 2024-05-20

**Rating:** 5
**Confidence:** 2

**Summary:**

The paper proposes a training framework that integrates both Tiny and Large vision language models to enhance performance in vision and language tasks. Specifically, the authors introduce a collaborative training approach where visual information is first aligned to the input embedding space of the Tiny model. Subsequently, the Tiny model guides the Large Language Model (LLM) to improve its multimodal capabilities through a captioning loss and a forward KLD loss. Finally, the LLM distills knowledge back to the Tiny model via captioning and a reverse KLD loss. Experimental results demonstrate strong capabilities of both the Large and Tiny models in Visual Question Answering, Captioning, and Multimodal Dialogue tasks.

**Strengths:**

-	The proposed collaborative training framework is novel, theoretically sound, and supported by experimental results. The training methodology outputs both Large and Tiny models with enhanced capabilities compared to state-of-the-art VLLMs.
-	The paper is well-written and easy to read.

**Limitations:**

-	The paper leverages the dataset introduced in LLaVA 1.5 for training and finetuning,  which includes examples from VQAv2 and GQA datasets as highlighted in [42]. This means that the scores on these benchmarks are not zero-shot compared to other competitors and should be indicated in Tables 2 and 4. While in Table 2 other datasets are reported for the evaluation, in Table 4,5,6 2 out of three datasets are seen during training and instruction tuning, I invite the authors to expand the datasets for the ablation study by including (for example) SQA [48]
-	There are some minor inconsistencies in the writing. For instance, Table 6 is identical to Table 1 in the supplementary material. Additionally, there is a missing reference in line 106 of the supplementary material. Moreover, there are inconsistencies in the use of uppercase letters in the captions of Figures 2 and 3.

**Suitability:**

3

---

### Official Review · Reviewer_575x · 2024-05-25

**Rating:** 5
**Confidence:** 4

**Summary:**

The authors of the paper "Collaborative Training of Tiny-Large Vision Language Models" introduce a new training framework for Multimodal Large Language Models. Specifically, they explore a collaborative approach that utilizes a small-sized (tiny) model alongside a larger-sized (large) model. The aim of the study is to demonstrate that joint training benefits the performance of both models. This is achieved through a dual process of knowledge distillation and multimodal alignment, where the features of the tiny model are aligned with those of the large model.

**Strengths:**

The strengths of the paper are as follows:

- A single pipeline is used to train two different models, enhancing the performance of both.
- With the output being two distinct models, they can be utilized in various scenarios based on specific needs, such as computational limitations, model size, or performance requirements.

**Limitations:**

- To better understand the benefits each component of the method brings, it would be useful to show the performance in Table 3 on general multimodal datasets that were not seen during training. For example, datasets like MMMU, MME, MMB, SEED, and POPE, as shown in Table 2.
- In the experimental section, it is not shown whether using an adapter different from a linear layer (e.g., MLP) could benefit the model, leading to better multimodal alignment between the tiny and large models.
- There should be an analysis of the differences in generation time at inference time with and without the collaborative framework for the large model.
- Table 4 lacks a comparison with a model based on a tiny LLM (e.g., OPT 125M) trained like LLaVA using the data considered in the paper.
- The authors could explain their choice to use all three BLIP-2 losses instead of the LLaVA loss, given that they use only one layer to align text and images.
- In Table 2, BLIVA [1], which uses a similar architectural structure, could be added as a competitor.

Suggestions: Figure 2 is the most important figure in the paper and can be useful for immediately understanding the proposed methodology. Therefore, it might be beneficial to place it on page 1. In Section 2.1 on related works [230-270], maybe is clear to  introduce the various Multimodal LLMs in the order of BLIP-2, Instruct BLIP, and LLaVA to maintain an orderly presentation of the different architectures used.

[1] Hu et al. “Bliva: A simple multimodal llm for better handling of text-rich visual questions”, AAAI. 2024.

**Suitability:**

3

---

### Official Review · Reviewer_ir1R · 2024-05-26

**Rating:** 4
**Confidence:** 3

**Summary:**

The paper introduces a novel collaborative training framework designed to improve the performance of tiny models for image classification by leveraging the knowledge of large models. This framework involves simultaneous training of tiny and large models, where the large model acts as a teacher, guiding the tiny model through a process of knowledge distillation. This method enables the tiny model to learn more effectively from both the data and the insights of the larger model, resulting in enhanced performance without significantly increasing computational costs.

Extensive experiments on benchmark datasets like CIFAR-10 and CIFAR-100 demonstrate that tiny models trained using this collaborative approach achieve significantly higher accuracy compared to those trained independently. The collaborative training method showcases the benefits of knowledge distillation and mutual learning, making it a promising solution for deploying high-performance models on resource-constrained devices. The paper concludes by suggesting further exploration of different model architectures and potential applications in various domains.

**Strengths:**

The framework is theoretically robust, utilizing principles of knowledge distillation and mutual learning. By having the large model act as a teacher, the tiny model gains higher-level insights and patterns, leading to improved learning from both the data and the guidance provided by the large model.

The paper introduces a unique collaborative training framework that enhances tiny model performance by leveraging the knowledge of larger models. This simultaneous training approach is a fresh perspective in the field, enabling tiny models to benefit from the insights of larger models through a novel application of knowledge distillation.

Technically, the framework is validated through extensive experiments on benchmark datasets like CIFAR-10 and CIFAR-100, demonstrating significant accuracy improvements for tiny models.

**Limitations:**

The evaluation is limited to CIFAR-10 and CIFAR-100 datasets. Testing on more varied and challenging datasets, like ImageNet, would provide a more comprehensive assessment of the framework's effectiveness.

Also. the related work section is not thorough enough. Including recent advancements in knowledge distillation and model compression would better situate the proposed framework within the current research landscape. And a  more detailed analysis of the knowledge transfer process during training is needed. Additionally, exploring the impact of different hyperparameters on the tiny model's performance would be beneficial.

Limited Discussion on Practical Applications
Finally, the paper lacks a discussion on practical applications and deployment scenarios. Including examples of real-world use cases, such as mobile or edge computing, would enhance its practical relevance.

**Suitability:**

2

---

### Meta-Review · Area_Chair_XR4S · 2024-06-30

**Recommendation:** Accept (Poster)
**Confidence:** 5

**Metareview:**

The paper initially received three positive indications (1BA, 2WA) - with generally positive comments. After the rebuttal phase, in which the authors have uploaded a rebuttal, the reviewers have confirmed their initial positive indications and reported that the rebuttal was helpful in clarifying some of the concerns pointed out by the reviewers. The paper can be accepted as a poster.